# Acoziborole resistance associated mutations in *Trypanosoma brucei* CPSF3

**Melanie Ridgway[1], Markéta Novotná[1], Cesar Mendoza-Martinez[1,2], Michele Tinti[1], Simone Altmann[1], Graeme Sloan[1,2], David Horn[1]***

**1** Faculty of Life Sciences, University of Dundee, Dundee, United Kingdom, **2** Drug Discovery Unit, Faculty of Life Sciences, University of Dundee, Dundee, United Kingdom

¤ Present address: Molecular and Biomedical Science, School of Biological Sciences, Adelaide University, Adelaide, South Australia, Australia

* d.horn@dundee.ac.uk

## Abstract

Acoziborole is a safe, single dose, oral therapy, for treatment of both early and late-stage sleeping sickness, a deadly disease caused by African trypanosomes. Other benzoxaboroles show efficacy against other trypanosomatids, apicomplexans, fungi, bacteria, and viruses. Acoziborole targets the trypanosome pre-mRNA processing endonuclease, cleavage and polyadenylation specificity factor 3 (CPSF3), and triggers CPSF3 degradation, but it remains unclear whether additional mechanisms contribute to efficacy. We used oligo targeting for site saturation mutagenesis of the native *CPSF3* gene. Among >1,500 edits around the putative drug binding site, only Asn[232]His edits conferred moderate resistance to acoziborole. Using a combinatorial oligo targeting method we edited multiple sites simultaneously, including sites that differ in human CPSF3, and found that an Asn[232]His, Tyr[383]Phe, Asn[448]Gln triple-mutant strain was >40-fold resistant to acoziborole. We used gene tagging to show that all three edits were on the same allele, and to show that triple-mutant CPSF3 was highly resistant to rapid acoziborole and proteasome-dependent degradation. Computational modelling revealed how the combinatorial mutations can disrupt acoziborole – CPSF3 interactions by introducing steric clash and by disrupting hydrophobic and water-mediated interactions. We conclude that acoziborole safety and efficacy can be explained by selective affinity for, and rapid turnover of, trypanosome CPSF3.

## Author summary

Diagnosis and treatment options, previously limited for sleeping sickness, have been transformed in recent years. Acoziborole, for example, is a new, safe, single dose, oral therapy for the treatment of this deadly disease. This drug can also be used without the need for cumbersome disease-stage diagnosis. Additional boron-based drugs also show great promise against a whole range

**Data availability statement:** The high-throughput sequencing data generated for this study have been deposited at the Sequence Read Archive under accession code PRJNA1365812 (https://www.ncbi.nlm.nih.gov/bioproject/PRJNA1365812). We also provide a Zenodo repository with details of homology models, representative MD clusters in pdb format, a CSV file with Free Energy and RMSD data, and a PDF file from the MD analysis: https://zenodo.org/records/17666549.

**Funding:** The work was supported by a Wellcome Centre Award (223608/Z/21/Z, D.H. was co-applicant), and a Wellcome Investigator Award (217105/Z/19/Z to D.H.). The funders had no role in study design, data collection and analysis, decision to publish, or preparation of the manuscript.

**Competing interests:** The authors have declared that no competing interests exist.

of other infectious diseases. Acoziborole targets an RNA processing enzyme in African trypanosomes, and triggers its degradation, but human cells express a similar enzyme, and alternative trypanosomal targets have also been suggested. Insights into how a drug interacts with its target can help to understand selective action against a pathogen, and to predict resistance, an ever-present threat for many drugs. We used a precision gene editing method to change the target protein in trypanosomes, editing single sites or multiple sites simultaneously. A triple-mutant was found to be both highly resistant to acoziborole and highly resistant to rapid degradation. Using computational models, we were able to explain how multiple mutations interfered with acoziborole binding to its target. The findings show how selective binding to a specific parasite enzyme makes acoziborole such a safe and effective drug.

## Introduction

Acoziborole (SCYX-7158/AN5568) and other benzoxaboroles that target CPSF3 have emerged as therapies or potential therapies for the treatment of sleeping sickness, nagana, Chagas' disease, leishmaniasis, malaria, toxoplasmosis, cryptosporidiosis, and cancer [1]. Benzoxaboroles also display antiviral, antibacterial, and antifungal activity. In the case of sleeping sickness, caused by African trypanosomes, acoziborole presents a key tool that could help meet and sustain the World Health Organization goal to interrupt disease transmission by 2030.

*Trypanosoma brucei gambiense* and *T. brucei rhodesiense,* responsible for human African trypanosomiasis, are closely related to *T. brucei brucei,* which causes nagana in cattle; all these parasites are transmitted by tsetse flies. Acoziborole is safe and >95% effective when administered to trypanosomiasis patients as a single oral dose of three 320 mg tablets [2,3]. This new therapy presents excellent new options for the treatment of both early infection and late-stage disease involving progression to the central nervous system, without the need for hospitalization, or painful and hazardous lumbar puncture for diagnosis and staging. Benzoxaboroles are also currently in trials against nagana in cattle [4] and are under development for the treatment of Chagas' disease [5], leishmaniasis [6,7], malaria [8] and cryptosporidiosis [9].

Acoziborole and other benzoxaboroles target CPSF3 in *T. brucei* [10–12], *Trypanosoma cruzi* [5,13], *Leishmania infantum* and *L. donovani* [6,7], *Plasmodium* [8], *Cryptosporidium* [9], and *Toxoplasma* [14,15]. It has been proposed that acoziborole does not bind CPSF3, however [16]. Notably, although a CPSF3 - $N^{232}H$ mutant was resistant to acoziborole, consistent with acoziborole – CPSF3 interaction, resistance was increased only moderately [12]. Acoziborole also triggers CPSF3 degradation in *T. brucei* [17], but it remains unclear whether additional mechanisms contribute to efficacy.

We have used oligo targeting for *CPSF3* gene editing in *T. b. brucei* and computational modelling to further explore acoziborole – CPSF3 interactions. Oligo targeting is a simple and DNA cleavage-free editing method that can deliver the full range of

possible base edits to native gene loci [18,19]. An oligo targeting screen of >1,500 CPSF3 edits for mutations that confer acoziborole resistance rediscovered the N[232]H mutant but failed to identify additional edits. Application of the approach in combinatorial mode, however, yielded a triple-mutant CPSF3 strain that was > 40-fold resistant to acoziborole, and CPSF3 that was highly resistant to acoziborole and proteasome-dependent degradation. Computational modelling further supported the view that acoziborole selectively engages *T. brucei* CPSF3 at the RNA substrate-binding pocket, thereby explaining the safety and efficacy of this treatment.

## Results

### CPSF3 mutagenesis and acoziborole resistance profiling

Gene editing using oligo targeting in *T. brucei* simply requires delivery of single-stranded oligodeoxynucleotides (ssODNs) by electroporation, using approx. 50 b 'reverse-strand' ssODNs [20]; a single allele is typically edited. We recently developed Multiplexed Oligo Targeting (MOT) library screening in *T. brucei* to examine a proteasomal drug target [18] and have now applied this approach to CPSF3 (Tb927.4.1340). Our previous analysis [12] revealed twenty-six amino acids located within 5 Å of the acoziborole binding site in CPSF3 (Fig 1A); with the cognate codons for these amino acids distributed over a region of approx. 1,200 bp in the *CPSF3* gene. Prior to assembling a mutant library, we determined the *CPSF3* allele replacement frequency using oligo targeting. Mismatch-repair supresses editing efficiency in *T. brucei* [20], and we found that this was also the case at the *CPSF3* locus. Using an ssODN to introduce a CPSF3 N[232]H edit, we obtained 6.5-times more acoziborole-resistant cells following transient *MSH2* knockdown for 24 h, with an estimated allele replacement frequency of approx. 0.01%. A similar mock assay but with no ssODN failed to yield any drug-resistant cells.

To assemble a CPSF3 mutant library, we designed twenty-six reverse-strand, 53-b, ssODNs, each with a centrally located degenerate 'NNN' (N = A, C, T, G) codon (S1 Table). Mismatch-repair was transiently knocked down for 24 h and each degenerate ssODN was delivered individually to avoid combinatorial editing in individual cells (Fig 1B), which could allow for the enrichment of bystander edits, complicating codon-based genotype to phenotype assessments when screening MOT libraries [18]. We estimated an average yield of approx. 325 codon edits per ssODN, which in the pooled library equated to approx. 6,500 edited cells among 50 million cells in a 150 ml culture volume; an average approx. 5-fold redundancy for each alternative codon. To assess editing at each of the targeted codons, we extracted genomic DNA before, and six hours after library assembly, PCR-amplified the edited region in the *CPSF3* gene, and deep-sequenced the amplicons (Fig 1B). A scan to quantify variant codons across the edited region revealed highly specific editing at all twenty-six targeted sites (Fig 1C). We concluded that the substantial majority of all 1,664 possible alternative codons encoding 520 CPSF3 variants were likely represented in our pooled MOT library.

The pooled MOT library was split to generate a pair of MOT libraries which were grown with acoziborole at 1 µM; approx. three times the EC$_{50}$ (Effective Concentration of drug to inhibit growth by 50%). The selected libraries were then split three ways and grown with acoziborole at either 3, 9 or 27 µM. In this case, only 3 µM selection yielded resistant cells. We then extracted genomic DNA from resistant cells, PCR-amplified the edited region in the *CPSF3* gene, deep-sequenced the products (Fig 1B), and quantified variant codons. The heatmap in Fig 1D shows relative representation of all 1,664 codon variants prior to drug-selection and following 1 or 3 µM acoziborole selection. Only the known N[232]H resistance-associated mutation [12,20], represented by both possible histidine codons, was enriched following selection; one by a single-nucleotide edit (*AAT - C*AT) and the other by a double-nucleotide edit (*AAT - CAC*).

### Triple-mutant CPSF3 is highly resistant to acoziborole

Several mutations in Apicomplexan CPSF3 have been linked to benzoxaborole resistance, including *Toxoplasma* Y[328]H, Y[483]N, S[519]C and E[545]K [14], and *Plasmodium* Y[408]S and D[470]N [8]. These sites are equivalent to sites targeted for editing above in *T. brucei*; N[232], Y[383], S[421], and N[448] (S1 Fig), with N[232]H already linked to acoziborole resistance in *T. brucei*

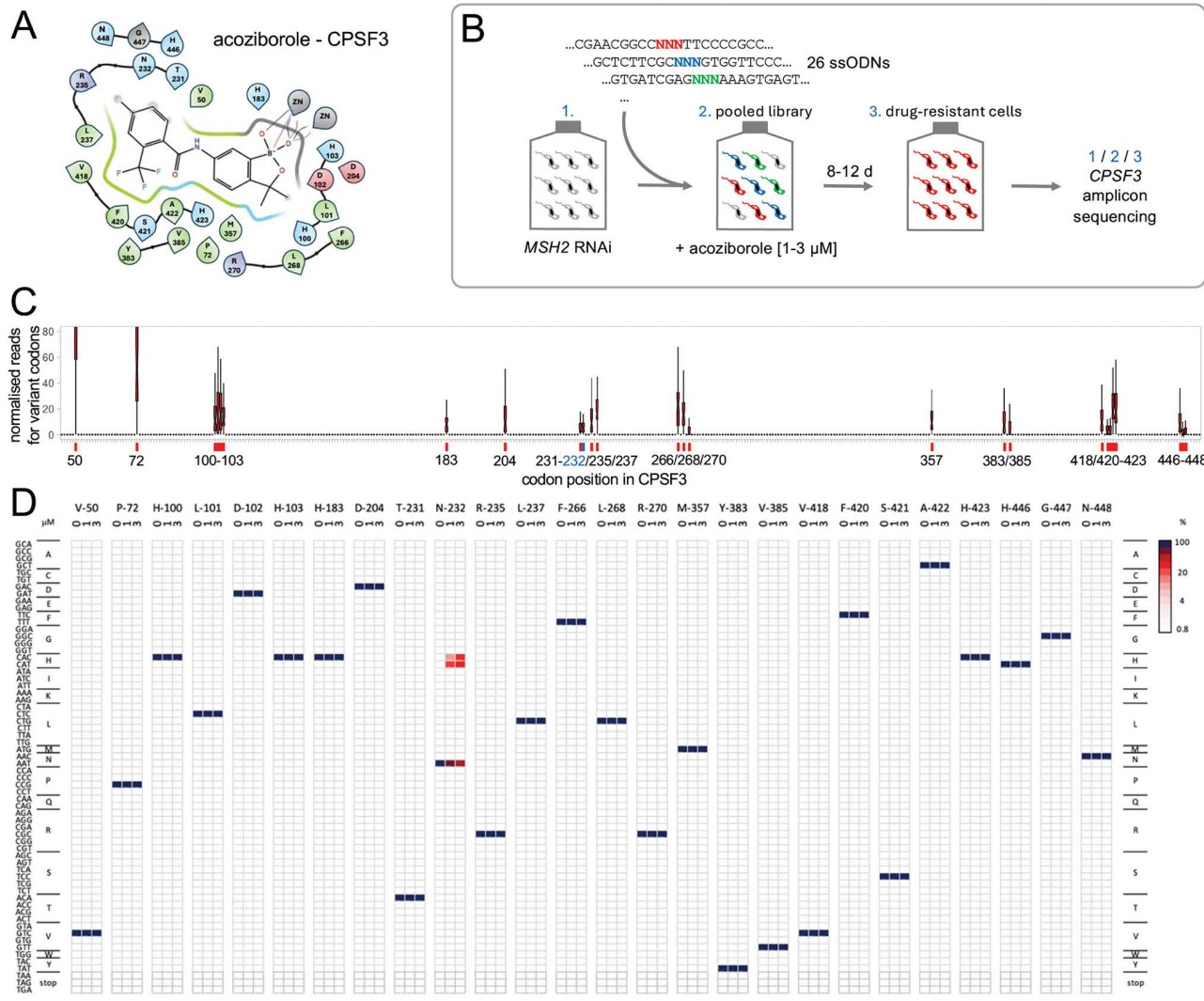

**Fig 1. CPSF3 mutagenesis and acoziborole resistance profiling. (A)** The ligand interaction diagram shows twenty-six *T. brucei* CPSF3 residues that are within 5Å of docked acoziborole. Green, hydrophobic; blue, positively charged; red, negatively charged; ZN, metal ions. **(B)** The schematic illustrates the MOT-library assembly and screening approach. Twenty-six single-stranded oligodeoxynucleotides (ssODN) with a central degenerate codon were individually transfected into *T. brucei MSH2* RNAi cells 24 h after inducing knockdown with tetracycline. Acoziborole selection was applied to the pooled library and cells from each step were subjected to *CPSF3* amplicon-sequencing. **(C)** The boxplot shows specific editing for all twenty-six targeted codons, indicated on the x-axis, and as determined by deep sequencing and codon variant scoring; average of >10 M reads mapped per site. Boxes indicate the interquartile range (IQR) and the whiskers show the range of values within 1.5 × IQR. **(D)** Codon variant scores for all sixty-four possible variants at all twenty-six targeted sites in the *CPSF3* gene are represented as a heatmap; for the unselected control sample, and for 1 or 3 µM acoziborole selection; average of >1.5 M reads mapped per site. Unedited codons are indicated (dark blue).

(see above). Pairs of co-transfected mutagenic ssODNs that target proximal sites are incorporated at high frequency in *T. brucei* [18] and a similar phenomenon is observed in yeast [19]. Given the paucity of acoziborole resistance associated mutations identified above, we exploited combinatorial oligo targeting to edit *CPSF3* and to seek to identify other sites that impact acoziborole efficacy.

We combined ssODNs designed to introduce an $N^{232}H^{CAT}$ edit, and three degenerate ssODNs designed to target $Y^{383}$, $S^{421}$ and $N^{448}$ for site saturation mutagenesis. Mismatch-repair was transiently knocked down for 24 h, five *T. brucei*

populations were transfected with the ssODN mix, pooled, split into two cultures, and grown with acoziborole at 3 µM. Resistant populations emerged, and ten sub-clones were assessed by Sanger sequencing, revealing three distinct combinatorial edits, all incorporating $N^{232}H^{CAT}$, as expected, and additionally with $Y^{383}F$ (either T*TC* or *TT*T) and $N^{448}Q$ or $N^{448}H$ (Fig 2A); all observed edits were heterozygous, while $S^{421}$ edits were not observed.

Acoziborole $EC_{50}$ dose response analysis was carried out for an $N^{232}H$ mutant and for a representative clone of each combinatorial mutant. The $N^{232}H$ mutant registered the expected moderate 3-fold increase in $EC_{50}$, but this was not significantly different to wild type cells ($P = 0.73$; one-way analysis of variance). In contrast, the $N^{232}H/ Y^{383}F$ (HFN), $N^{232}H/ Y^{383}F/ N^{448}H$ (HFH) and $N^{232}H/ Y^{383}F/ N^{448}Q$ (HFQ) mutants all registered a significantly different $EC_{50}$ ($P < 0.0001$) relative to wild type cells (Fig 2B). In particular, the HFQ triple mutant registered >40-fold increased $EC_{50}$ suggesting synergy between the $N^{232}H$ and $Y^{383}F$ edits and additional synergy with the $N^{448}Q$ edit.

### Triple-mutant CPSF3 resists acoziborole-induced turnover

Recovery of triple-edited cells using the combination of ssODNs detailed above suggested remarkably efficient incorporation of multiple ssODNs, targeting sites 651 bp apart, likely at a common DNA replication or transcription fork. At this point, it remained unclear as to whether the heterozygous combinatorial edits were present on the same *CPSF3* allele, however. To determine whether this was indeed the case, we engineered HFQ triple mutant strains to express a native *CPSF3*

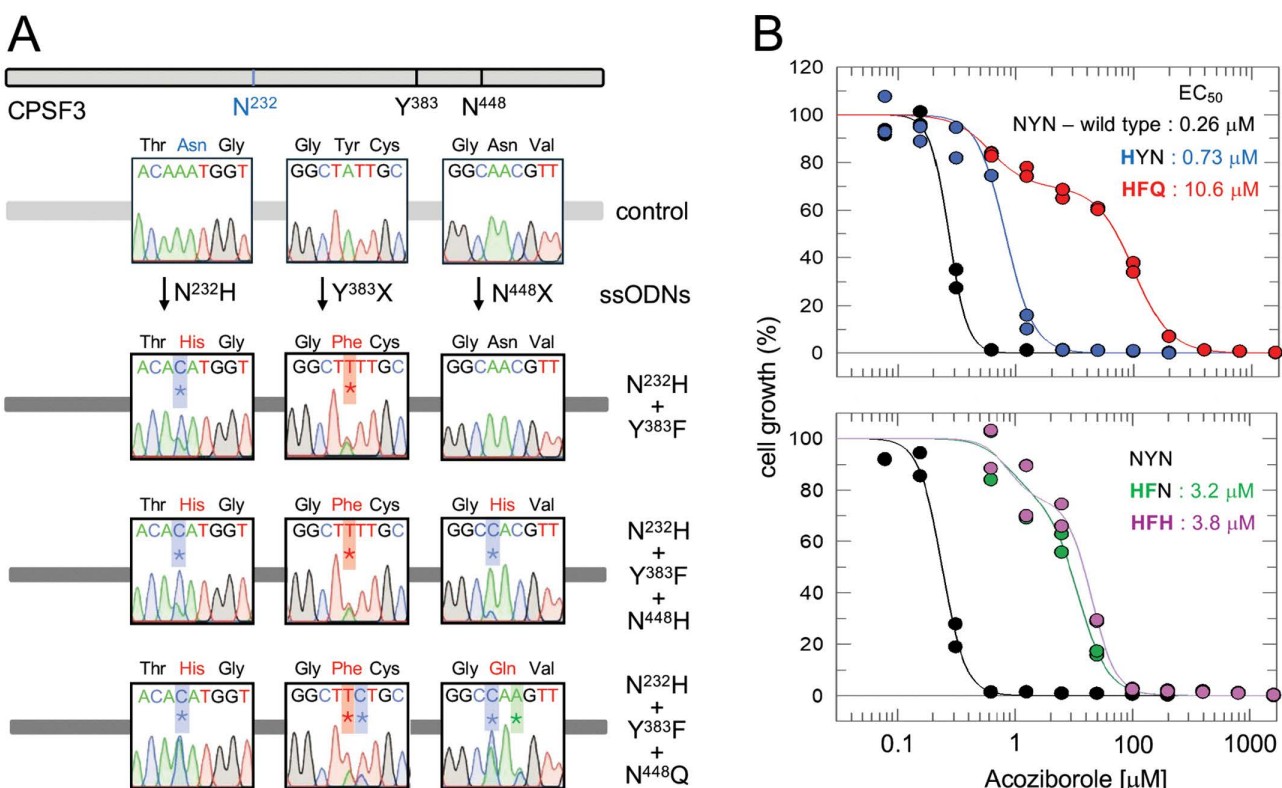

**Fig 2. Triple-mutant CPSF3 cells are highly resistant to acoziborole. (A)** Co-editing at adjacent sites following delivery of multiple ssODNs. The schematic representation of the *CPSF3* sequence highlights residues successfully targeted for editing. Sequence traces show the outcomes following combinatorial ssODN delivery and selection with 3 µM acoziborole. Edited nucleotides are marked by asterisks. **(B)** Dose-response curves and $EC_{50}$ values for the mutants. All acoziborole dose responses were measured in duplicate and repeated 3-4 times and for 1-2 clones for each mutant. Representative dose-response curves are shown.

allele fused to green fluorescent protein [12]. Amplification and Sanger-sequencing of the *CPSF3*GFP alleles from a series of clones revealed either wild-type sequence or triple-mutant sequence, indicating tagging of either allele, and demonstrating the presence of all three edits on the same allele (Fig 3A).

We next asked whether acoziborole-resistant triple mutant CPSF3 was resistant to acoziborole-induced turnover [17]. We analysed a pair of acoziborole resistant CPSF3GFP strains, one with the triple-mutant *HFQ* allele tagged with GFP and another with the wild-type *NYN* allele tagged with GFP. We treated these cells with 3, 6 or 10 μM acoziborole for 30 minutes and assessed the samples by protein blotting. The wild-type protein was rapidly turned over by acoziborole, and pre-treatment with the proteasome inhibitor MG132 blocked turnover (Fig 3B), confirming that NYN-CPSF3GFP turnover was proteasome-dependent [17]. In contrast, mutant HFQ-CPSF3GFP was resistant to turnover (Fig 3B). We concluded that triple mutant CPSF3 resisted rapid acoziborole and proteasome-dependent turnover.

## Computational modelling and acoziborole docking with CPSF3 mutants

Identification of multiple resistance-associated mutations around the acoziborole binding site, which is also part of the RNA substrate binding site, provided compelling evidence for CPSF3 binding by this small molecule. To further explore interactions between acoziborole and CPSF3, we modelled and compared wild-type NYN-CPSF3 and multiple CPSF3 mutants. We used crystal structures for related RNAse's as templates and generated *T. brucei* CPSF3 homology models docked with acoziborole. To select a binding pose, we prepared single mutants for all possible residues at $N^{232}$, $Y^{383}$, and $N^{448}$, and used ΔΔG Molecular Mechanics General Born Surface Area (MM/GBSA) free energy calculations to track those mutations [21]. Among the ensemble of predicted poses, we selected pose two derived from the 3IEM-based homology model, which exhibited sensitivity to mutations at all three sites (S2 Fig).

In the selected pose, CPSF3 $N^{232}$, $Y^{383}$ and $N^{448}$ surround the acoziborole binding site, with distances from the ligand of 2.17, 3.67 and 5.45 Å, respectively. Mutations at these sites may have steric effects or impact acoziborole affinity. To explore these potential impacts, we ran molecular dynamics simulations, which suggested interactions with $N^{232}$ and $Y^{383}$; via a water network and a hydrophobic interaction, respectively (Fig 4A). Simulations were then clustered by Root Mean

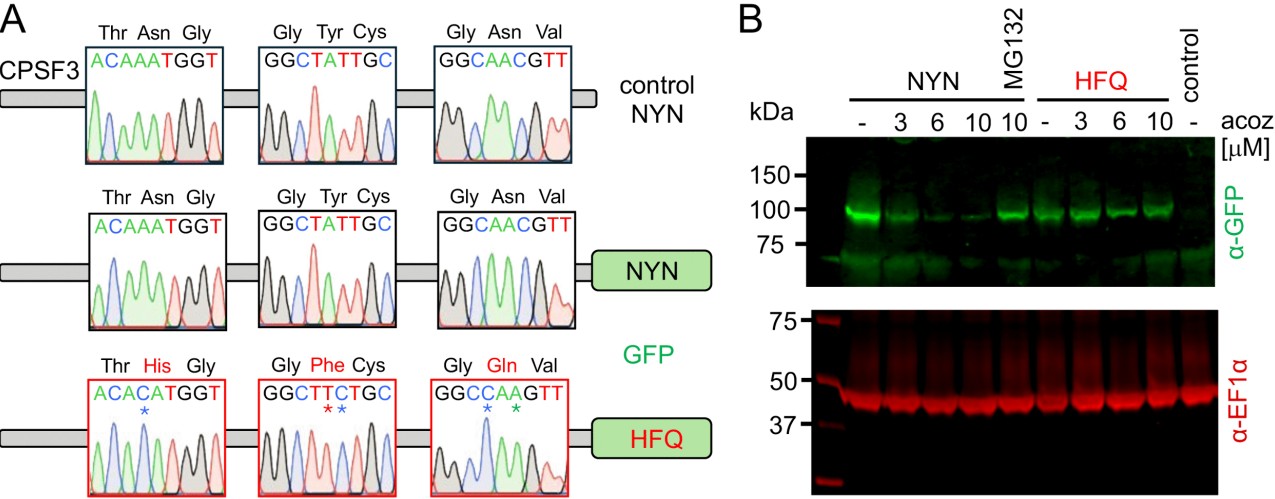

**Fig 3. Triple-mutant CPSF3 resists acoziborole-induced turnover. (A)** GFP-tagged alleles in the triple-edited 'HFQ' strain were amplified, sequenced and compared to a control 'NYN' sequence. Sequence traces show alternative tagged alleles in the 'HFQ' strain; an NYN allele and an HFQ allele. **(B)** The protein blot shows CPSF3GFP expression in acoziborole-resistant cells with either the NYN or HFQ allele tagged, and following acoziborole exposure for 30 min. EF1-α serves as a loading control. Cells lacking a GFP-tag, and cells with the NYN allele tagged with GFP, that were pre-exposed to the proteasome inhibitor MG132, are included as controls.

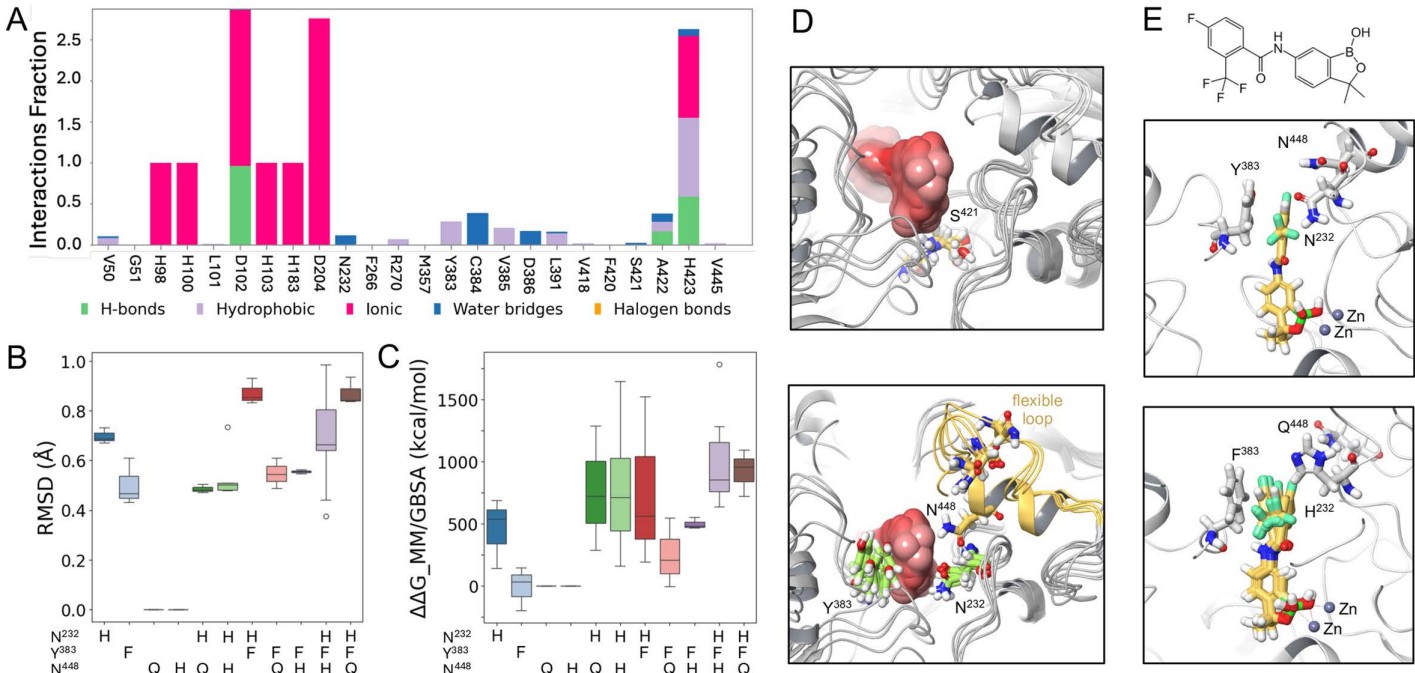

**Fig 4. Computational modelling and acoziborole docking with mutant CPSF3. (A)** Acoziborole interaction fractions observed for the CPSF3 residues indicated during Molecular Dynamics (MD) simulation. **(B)** Root Mean Square Deviation (RMSD) values following the mutations indicated, the reference is wild type in all cases. **(C)** ΔΔG MM/GBSA free energy value changes following the mutations indicated, the reference is wild type in all cases. **(D)** Ensembles of conformations extracted from the MD trajectory highlighting S421 (upper panel) and N448 associated with a flexible loop (lower panel). **(E)** Binding poses proposed for acoziborole in the CPSF3 homology model (upper panel) and in the 'NYN' triple mutant (lower panel). The N/H232, Y/F383, and N/Q448 residues and Zinc ions are indicated.

Square Deviation (RMSD), and three representative snapshots were taken for mutation analysis. We created single, double, and triple mutant combinations of N232H, Y383F, N448Q, and N448H, and determined the ΔΔG MM/GBSA associated with each mutant. Upon mutation, measures of RMSD indicate a number of steric effects. Notably, the four mutants we describe above (Fig 2) register the greatest increase in RMSD, which is particularly pronounced in the 'HF' double mutant and in the 'HFQ' triple mutant (Fig 4B), consistent with our *in cellulo* experimental data. Finally, free energy (ΔΔG MM/GBSA) was predicted to be increased for multiple double mutants compared with single mutants and further increased for triple mutants compared with double mutants (Fig 4C).

Our predictions also potentially explain why we failed to recover mutations at S421 above; because the side chain is not predicted to directly face the ligand (Fig 4D, upper panel). We can also see why N448 does not appear in the molecular dynamics interaction network (Fig 4A). Although N448 appears in close proximity to the ligand in the initial structural model, this proximity is predicted to be transient due to a highly flexible and dynamic loop (Fig 4D, lower panel). While mutation at N448 alone is predicted to have no impact on RMSD (Fig 4B) or free energy (Fig 4C), mutation at N448 is predicted to impact free energy in a synergistic manner with N232H and Y383F (Fig 4C), suggesting that these additional mutations impact the flexible loop. This is also consistent with recovery of N448 mutations only in the context of triple mutants above (Fig 2A). We conclude that the predictions were in agreement with our experimental data, which suggested synergy between the resistance-associated N232H and Y383F edits and additional synergy with the N448Q edit (Fig 2B).

Thus, *in silico* modelling and molecular dynamics simulations suggested that triple-mutant CPSF3 was resistant to acoziborole due to both steric clash between the ligand and the mutated residues, and disruption of hydrophobic and

water-mediated interactions, thereby reducing affinity. The poses shown in Fig 4E indicate how the $N^{232}$, $Y^{383}$, and $N^{448}$ residues cluster around the fluorinated benzene ring in acoziborole (upper panel) and how acoziborole stability is impacted in the HFQ triple-mutant (lower panel).

## Discussion

Acoziborole targets trypanosome CPSF3 and displays excellent anti-parasite efficacy and low host toxicity. CPSF3-independent modes of action have been proposed for acoziborole, however, and only a single resistance-associated CPSF3 mutation has been described. Guided by a CPSF3 homology model, we previously identified twenty-six residues within 5Å of docked acoziborole [12]. We targeted all twenty-six of these residues for site saturation mutagenesis using oligo targeting, and constructed and screened libraries of *T. brucei* cells with mutations at the otherwise native *CPSF3* locus. Following drug selection, deep sequencing and codon variant scoring, only the previously known drug resistance mutation was identified, $N^{232}H$. We subsequently used a combinatorial oligo targeting approach to edit multiple sites simultaneously, and found that an $N^{232}H$, $Y^{383}F$, $N^{448}Q$ triple-mutant strain was > 40-fold resistant to acoziborole. Triple-mutant CPSF3 was also resistant to acoziborole-dependent turnover. Finally, computational modelling predicted contributions from both steric effects and reduced affinity as explanations for increased acoziborole-resistance observed for the CPSF3 triple mutant.

Our gene editing screen using oligo targeting suggested highly restrictive mutational acoziborole-resistance space in trypanosome CPSF3, which contrasts to forty-six distinct drug-resistance associated mutations identified recently in the *T. brucei* proteasome β5 subunit using a similar approach [18]. The edits that failed to yield acoziborole resistant cells may be explained by either failure to impact drug binding, or simply because they yield defective CPSF3; perhaps because the drug-binding pocket in CPSF3 is highly constrained to maintain RNA-binding. Regardless, our study demonstrates limited scope for spontaneous mutation leading to acoziborole resistance in single-nucleotide accessible space in the substrate and drug-binding pocket. Indeed, the moderate shift in resistance observed due to $N^{232}H$ mutation may be insufficient to lead to treatment failure in a clinical setting, while the $N^{448}Q$ mutation requires a double-nucleotide change, suggesting that acoziborole could be a durable monotherapy. A recently reported CRISPR diagnostic tool presents an excellent opportunity for surveillance in this regard [22]. The $N^{232}$ and $N^{448}$ positions in the trypanosome protein are occupied by Y and E at the equivalent positions in the human protein. Although the acoziborole–resistance-associated $N^{232}H$ and $N^{448}Q$ mutations we observe at these positions do not directly mimic the human sequence, these differences between human and trypanosome CPSF3 likely help to explain why acoziborole is a safe therapy.

Combinatorial oligo targeting proved to be effective for editing *T. brucei CPSF3* and could be used to assess other drug targets in trypanosomatids. Although insights from Apicomplexan parasites guided our analysis here in terms of targeting $Y^{383}$ and $N^{448}$, the mutations we observe at these sites are distinct from those drug-resistance associated mutations previously observed in Apicomplexans; $Y^{383}F$ in *T. brucei* as opposed to $Y^{483}N$ in *Toxoplasma* and $Y^{408}S$ in *Plasmodium*, and $N^{448}H$ or $N^{448}Q$ in *T. brucei* as opposed to $E^{545}K$ in *Toxoplasma* and $D^{470}N$ in *Plasmodium*. This was possible because we used degenerate ssODNs to target $Y^{383}$ and $Y^{448}$ for site saturation mutagenesis in *T. brucei*. Thus, known drug-resistance associated mutations in Apicomplexans guided this study, but pools of degenerate ssODNs used in combinatorial oligo targeting format could be informative even without such prior knowledge.

Acoziborole was recently shown to target *T. brucei* CPSF3 for protein turnover [17], and we found that the 'HFQ' triple-mutant was highly resistant to acoziborole and proteasome-dependent turnover. Notably, we also found that the triple-mutant strain yielded biphasic dose response curves, and we suggest that this was due to the combined impacts of acoziborole on inhibiting CPSF3 activity and on CPSF3 turnover. Although we cannot rule out interaction with a second target, we suggest that acoziborole both inhibits activity and promotes turnover by binding the same site on CPSF3. Our CPSF3 homology models and molecular dynamics simulations suggest how the $N^{232}$, $Y^{383}$, and $N^{448}$ residues cluster around the fluorinated benzene ring in acoziborole and how acoziborole – CPSF3 interactions are disrupted in the HFQ

triple-mutant. We conclude that acoziborole safety and efficacy can be explained by selective affinity for, and turnover of, trypanosome CPSF3.

## Materials and methods

### *T. brucei* growth and manipulation

Bloodstream form Lister 427 *T. brucei* wild-type and *MSH2* RNAi [20] strains were cultured in HMI-11 (Gibco) supplemented with 10% fetal bovine serum (Sigma) at 37°C and with 5% $CO_2$ in a humidified incubator. Tetracycline inducible 2T1-RPa[i]MSH2 RNAi cells were generated previously [20]. Genetic manipulation was carried out using electroporation with a Nucleofector (Lonza), and a human T-cell kit (Lonza), with the Nucleofector set to Z-001 (Amaxa). To calculate allele replacement frequency, transient *MSH2* knockdown was induced 24 h prior to transfection by the addition of 1 µg/mL tetracycline (Sigma). Acoziborole (MedChem Express) was then added 6 h after transfection of an ssODN designed to introduce a CPSF3 N[232]H edit and cells were serially diluted in 96-well plates. Number of resistant clones was determined 5 days later considering 50% survival following transfection. GFP-tagged CPSF3 was expressed in a wild-type *T. brucei* background as described previously by Wall *et al.* [12].

### Assembly and screening of multiplexed oligo targeting libraries

For site saturation mutagenesis, we used a set of twenty-six degenerate single-stranded oligodeoxynucleotides (ssODNs, Thermo Fisher Scientific), as described [20]. Briefly, we used 40 µg of each ssODNs in 10 µl of 10 mM Tris-HCl, pH 8.5, mixed with 5 million *T. brucei* 'inducible MSH2 RNAi' cells in 100 µl transfection buffer. *MSH2* knockdown was induced 24 h prior to transfection. Each ssODN was transfected individually and the cells were then pooled in 150 ml of medium to generate the library. After 6 h, cells from 50 ml of the culture were cryo-preserved, cells from another 25 ml were collected for (pre-selection) DNA extraction, and the remainder were split into two cultures and subjected to selection with 1 µM acoziborole in a total volume of 75 ml per culture. Following eight or ten days, 50 ml of each culture was collected for DNA extraction, and the remainder was split into three cultures that were subjected to selection with either 3 µM, 9 µM or 27 µM acoziborole. Only 3 µM selection yielded resistant cells, and these cultures were collected for DNA extraction. Genomic DNA was extracted using a Qiagen DNeasy Kit followed by PCR amplification of the *CPSF3* gene using Q5 DNA Polymerase (NEB). PCR products were purified using a PCR Purification Kit (Qiagen).

### Amplicon sequencing and codon variant scoring

To identify edits, *CPSF3* amplicons were subjected to deep sequencing using DNBSEQ (BGI Genomics), as described [18]. Briefly, filtering of sequencing reads was performed using SOAPnuke and codon variant scoring was performed with the OligoSeeker (0.0.5) Python package [23]. To visualise codon variant scores, we performed a normalization step by dividing each codon variant score by the total reads for that position and converted the fraction of reads to a percentage. This was followed by background correction whereby we subtracted the values for the control sample. Negative values were replaced with 0, and average values for the duplicate libraries were calculated to give codon variant scores. Edits for codons registering >100 read-counts in the control sample, including all single nucleotide variants, were excluded from the analyses shown in Fig 1C.

### Combinatorial oligo targeting

Oligo targeting was carried out using ssODNs essentially as described above, but in five batches with 10 µg of each ssODNs mixed with 10 million *T. brucei* 'inducible MSH2 RNAi' cells in 100 µl transfection buffer. The batches were pooled and selection with 3 µM acoziborole was applied 6 h later. Genomic DNA extraction and PCR amplification were carried out as above and, to identify edits, amplicons were subjected to Sanger sequencing (Genewiz, Azenta Life Sciences).

## Dose-response assays

To determine the Effective Concentration of drug to inhibit growth by 50% ($EC_{50}$), cells were plated in 96-well plates at 1 x $10^3$ cells/ml in a 2-fold serial dilution of selective drug. Plates were incubated at 37°C for 72 h, 20 µl resazurin sodium salt (AlamarBlue, Sigma) at 0.49 mM in PBS was then added to each well, and plates were incubated for a further 6 h. Fluorescence was determined using an Infinite 200 pro plate reader (Tecan) at an excitation wavelength of 540 nm and an emission wavelength of 590 nm. $EC_{50}$ values were derived using Prism (GraphPad), or Graphit in the case of biphasic dose response curves.

## Protein blotting

Twenty-five million *T. brucei* cells were treated with acoziborole for 30 min and collected by centrifugation. The proteasome inhibitor MG132 was added to a culture at 20 µM, 30 mins prior to acoziborole treatment. RIPA buffer and proteinase inhibitors were added, and samples sonicated for 10 cycles (30 sec on/off), centrifuged, and the supernatant collected. Proteins were resolved using SDS-PAGE in 4–12% Bis-Tris gels (Invitrogen) and then transferred to PVDF membrane using the iBlot 2 system (Invitrogen). Membranes were blocked with blocking buffer (50 mM Tris-HCl pH 7.4, 0.15 M NaCl, 0.25% BSA, 0.05% (w/v) Tween-20, and 2% (w/v) fish skin gelatin). The following antibodies were used in blocking buffer: rabbit α-GFP (Invitrogen, A-11122, 1:2000) and mouse α-EF1α (Millipore, 1:10000) overnight at 4°C. Secondary antibodies were α-rabbit IRDye800 and α-mouse IRDye680 (1:15000 and 1:10000, respectively, LI-COR) for 1 h at room temperature. Blots were analysed using the LI-COR Odyssey CLx Imager and Image Studio 6.0.

## Homology modelling, docking and molecular dynamics simulations

*T. brucei* CPSF3 homology models were generated using the crystal structures with PDB ID: 3IEM (*Thermus thermophilus HB8*) [24], 6Q55 (*Cryptosporidium hominis*) [9], and 8T1Q (*Homo sapiens*) [25] as templates, and the homology modelling tool implemented in Maestro (Schrodinger inc.). Acoziborole was docked using Glide XP [26], and the top two poses for each model were selected for further analysis. Two docking poses were also generated using Boltz2, a state-of-the-art generative model for protein–ligand docking [27]. The ligand and protein complex was placed in a cubic box of water and 0.15 M NaCl at 27 °C. We ran 500 ns of molecular dynamics simulations following a multistage equilibration protocol designed to gradually relax positional restraints and achieve thermodynamic stability of the system before production sampling. Initially, the system was constructed and subjected to energy minimisation to remove steric clashes and optimise atomic geometries. This was followed by a short Brownian dynamics simulation in the canonical (NVT) ensemble at 10 K for 100 ps, using small integration time steps. During this simulation, positional restraints were applied to all solute heavy atoms to allow the solvent and ions to equilibrate around the solute. The simulation was then continued under NVT conditions for an additional 12 ps with the same restraints to ensure complete thermal equilibration of the solvent environment. Subsequent equilibration was performed under NPT conditions at 10 K and 1 bar for 12 ps with solute restraints to allow the system density to adjust. A further 12 ps NPT simulation was carried out under the same conditions to ensure pressure stability. The positional restraints were then released, and the system was equilibrated for 24 ps under NPT conditions to allow for the complete relaxation of all atoms. Finally, an unrestrained production simulation was performed under stable NPT conditions for 500 ns to collect trajectory data for analysis. Post-simulation analyses were conducted on the resulting trajectories to evaluate structural stability, conformational changes, and energetic properties throughout the simulation. We determined the ΔΔG MM/GBSA associated with the change. In the case of histidine, we analysed all the possible protonation states. We tracked the movement of the ligand in the binding site using RMSD, employing an MCS protocol on the ligand.

## Supporting information

**S1 Fig. The protein sequence alignment shows the region of *T. brucei* CPSF3 (Tb927.4.1340) that includes the mutations in the acoziborole resistant triple-mutant (green background).** Other sites assessed by oligo targeting are shown in red text. Sites associated with benzoxaborole resistance in Apicomplexan parasites (*Plasmodium falciparum*, PF3D7_1438500, Y[408]S and D[470]N; Toxoplasma gondii, TGME49_285200, Y[328]H, Y[483]N, S[519]C and E[545]K) and noted in the main text are shown in light blue text. The *Thermus thermophilus* protein used as template for homology modelling (BAD70075) and the human CPSF3 (AAF00224.1) are also shown for reference.
(PDF)

**S2 Fig. Computational modelling of acoziborole ligand affinity following mutation at the sites indicated in the CPSF3 homology models.** Pose 2, for the 3IEM-based model has the greatest impact following mutations at these sites.
(PDF)

**S1 Table. Oligonucleotides used in this study.** The set of twenty-six single-stranded oligodeoxynucleotides used for oligo targeting are shown. Also shown are the primers used to generate the CPSF3 amplicon, and the primer used for Sanger sequencing.
(PDF)

## Acknowledgments

We thank Mark C Field (University of Dundee) and Martin Zoltner (Charles University, Prague) for discussions on acoziborole-induced CPSF3 turnover, and Lindsay Tulloch (University of Dundee) for assistance with formatting $EC_{50}$ data using Graphit.

## Author contributions

**Conceptualization:** Melanie Ridgway, David Horn.

**Data curation:** Michele Tinti.

**Formal analysis:** Melanie Ridgway, Cesar Mendoza-Martinez, David Horn.

**Funding acquisition:** David Horn.

**Investigation:** Melanie Ridgway, Markéta Novotná, Simone Altmann.

**Project administration:** David Horn.

**Supervision:** Graeme Sloan, David Horn.

**Validation:** Cesar Mendoza-Martinez.

**Visualization:** Cesar Mendoza-Martinez.

**Writing – original draft:** Melanie Ridgway, Cesar Mendoza-Martinez, David Horn.

**Writing – review & editing:** Melanie Ridgway, Markéta Novotná, Cesar Mendoza-Martinez, Michele Tinti, Simone Altmann, David Horn.

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
