## [Decision Letter · Decision Letter 0]

8 Feb 2026

PPATHOGENS-D-25-02987

Acoziborole resistance associated mutations in trypanosome CPSF3

PLOS Pathogens

Dear Dr. Horn,

Thank you for submitting your manuscript to PLOS Pathogens. After careful consideration, we feel that it has merit but does not fully meet PLOS Pathogens's publication criteria as it currently stands. Therefore, we invite you to submit a revised version of the manuscript that addresses the points raised during the review process.

We look forward to receiving your revised manuscript.

Kind regards,

Christine Clayton

Academic Editor

PLOS Pathogens

Margaret Phillips

Section Editor

PLOS Pathogens

Sumita Bhaduri-McIntosh

Editor-in-Chief

PLOS Pathogens

orcid.org/0000-0003-2946-9497

Michael Malim

Editor-in-Chief

PLOS Pathogens

orcid.org/0000-0002-7699-2064

**Additional Editor Comments:**

Your paper has been reviewed by one expert referee and I have read it myself. Since it is a straightforward paper with clear results, the expert suggests only minor modifications - including fewer claims of "novelty" and clearer acknowledgement that your mutant design was reliant on previous results from other organisms. I agree with all of the reviewer's suggestions.

**Journal Requirements:**

At this stage, the following Authors/Authors require contributions: Melanie Ridgway, Marketa Novotna, Cesar Mendoza-Martinez, Michele Tinti, Simone Altmann, Graeme Sloan, and David Horn. Please ensure that the full contributions of each author are acknowledged in the "Add/Edit/Remove Authors" section of our submission form.

4)

Some material included in your submission may be copyrighted. According to PLOS’s copyright policy, authors who use figures or other material (e.g., graphics, clipart, maps) from another author or copyright holder must demonstrate or obtain permission to publish this material under the Creative Commons Attribution 4.0 International (CC BY 4.0) License used by PLOS journals. Please closely review the details of PLOS’s copyright requirements here: PLOS Licenses and Copyright. If you need to request permissions from a copyright holder, you may use PLOS's Copyright Content Permission form.

Potential Copyright Issues:

i) Figure 1B: Please confirm whether you drew the images / clip-art within the figure panels by hand. If you did not draw the images, please provide (a) a link to the source of the images or icons and their license / terms of use; or (b) written permission from the copyright holder to publish the images or icons under our CC BY 4.0 license. Alternatively, you may replace the images with open source alternatives. See these open source resources you may use to replace images / clip-art:

- https://openclipart.org/

**Reviewers' Comments:**

Reviewer's Responses to Questions

**Part I - Summary**

Reviewer #1: This is a very clean study identifying mutations that could impact the activity of acoziborole in T. brucei. The work is technically well-performed and the manuscript well-structured and written. Overall, it is very nice, although the results are not particularly remarkable (but few really are).

**Part II – Major Issues: Key Experiments Required for Acceptance**

Reviewer #1: none

**Part III – Minor Issues: Editorial and Data Presentation Modifications**

Reviewer #1: With the exception of 1 and 2, these are just suggestions with respect to emphasis - things the authors might consider

1. The titles refer to "trypanosome" CPSF3 when only the T. brucei CPSF3 was studied. The results may not apply to other trypanosomes

2. The authors indicate that the approach to making triple mutants was "novel" in several places - which strictly speaking might be true but is perhaps a bit of an oversell. It was an obvious approach. By this criterion, every new experiment/approach that had not been explicitly performed previously is "novel"

3. a little more discussion of why they think the S412 mutant did not confer any detectable resistance might be useful.

4. While the authors don't hide the fact, they also don't emphasize that making the triple mutant was only possible because single mutations in these sites had been discovered in other organisms - but where not discovered using their approach in T. brucei. That is, without these other mutations to test, this paper largely wouldn't exist (the N232 mutant was reported in the Altmann 2022 paper). This represents a potential limitation of this experimental approach (it is rare to have other drug resistance info from so many other organisms in response to the same class of drugs).

5. The difficulty in inducing and selecting for resistance to this class of compounds in trypanosomes is pretty remarkable. The authors note this, but this point feels quite undersold - and I think more comment in this regard would emphasize better the impact of this study (which otherwise is modest).

PLOS authors have the option to publish the peer review history of their article (what does this mean? ). If published, this will include your full peer review and any attached files.

**Do you want your identity to be public for this peer review?** For information about this choice, including consent withdrawal, please see our Privacy Policy .

Reviewer #1: **Yes:** Rick L. Tarleton

**Figure resubmission:**
---

## [Editor Report · Decision Letter 1]

23 Feb 2026

Dear Dr. Horn,

We are pleased to inform you that your manuscript 'Acoziborole resistance associated mutations in Trypanosoma brucei CPSF3' has been provisionally accepted for publication in PLOS Pathogens.

Best regards,

Christine Clayton

Academic Editor

PLOS Pathogens

Margaret Phillips

Section Editor

PLOS Pathogens

Sumita Bhaduri-McIntosh

Editor-in-Chief

PLOS Pathogens

orcid.org/0000-0003-2946-9497

Michael Malim

Editor-in-Chief

PLOS Pathogens

orcid.org/0000-0002-7699-2064
---

## [Editor Report · Acceptance letter]

Dear Dr. Horn,

We are delighted to inform you that your manuscript, "Acoziborole resistance associated mutations in Trypanosoma brucei CPSF3," has been formally accepted for publication in PLOS Pathogens.

Best regards,

Sumita Bhaduri-McIntosh

Editor-in-Chief

PLOS Pathogens

orcid.org/0000-0003-2946-9497

Michael Malim

Editor-in-Chief

PLOS Pathogens

orcid.org/0000-0002-7699-2064